# On Column Binding a Real-Time Biosensor for *β*-lactam Antibiotics Quantification

**DOI:** 10.3390/molecules25051248

**Published:** 2020-03-10

**Authors:** Shahla M. Abdullah, Shwan Rachid

**Affiliations:** 1Medical Laboratory Science Department, College of Science, University of Raparin, Ranyia 46012, Sulaymaniyah, Iraq; shahla.mohammed@uor.edu.krd; 2Biology Department, Faculty of Science and Health, Koya University, Koysanjaq 44023, Erbil, Iraq; 3Charmo Center for Research, Training, and Consultancy, Charmo University, Chamchamal 46023, Sulaymaniyah, Iraq

**Keywords:** biosensor, molecular quantification, penicillin-binding protein, penicillin, Bocillin FL, antibiotics

## Abstract

This work aimed to develop accurate, quick, and practical tools for the detection of residues of penicillin G antibiotic in biological and non-biological samples. The assays were developed based on the binding mechanism of *β*-lactam to penicillin-binding proteins; samples of different concentrations of penicillin G were incubated with in vitro expressed 6X-Histidine-tagged soluble penicillin-binding protein (PBP2x*) of *Streptococcus pneumoniae* (*S. pneumoniae*), whereby penicillin G in samples specifically binds to PBP2x*. The fluorescent-labeled *β*-lactam analogue Bocillin FL was used as a competent substrate, and two different routes estimated the amounts of the penicillin G. The first route was established based on the differences in the concentration of non-bounded Bocillin FL molecules within the reactions while using a real-time polymerase chain reaction (PCR)-based method for fluorescence detection. The second route depended on the amount of the relative intensity of Bocillin FL bounded to Soluble PBP-2x*, being run on sodium dodecyl sulfate–polyacrylamide gel electrophoresis (SDS-page), visualized by a ChemiDoc-It^®^2 Imager, and quantified based on the fluorescence affinity of the competent substrate. While both of the methods gave a broad range of linearity and high sensitivity, the on column based real-time method is fast, non-time consuming, and highly sensitive. The method identified traces of antibiotic in the range 0.01–0.2 nM in addition to higher accuracy in comparison to the SDS-based detection method, while the sensitivity of the SDS-based method ranged between 0.015 and 2 µM). Thus, the on column based real time assay is a fast novel method, which was developed for the first time based on the binding inhibition of a fluorescence competitor material and it can be adapted to screen traces of penicillin G in any biological and environmental samples.

## 1. Introduction

*β*-lactam antibiotics, which include penicillin, cephalosporin, monobactams, and carbapenems, are among the most widely used antibiotics and they represent a highly successful and effective treatment [1,2,3] in veterinary medicine used in the treatment of septicemia, urinary and pulmonary [4,5] conditions, and in mastitis treatment courses [6,7,8]; also, they have been used to boost animal growth [2]. The maximum residue limits were set at 4 µg L^−1^ for penicillin G, penethamate (penicillin G), amoxicillin and ampicillin (penicillins A), and at 30 µg L^−1^ for cloxacillin, oxacillin, dicloxacillin, and nafcillin in milk (penicillins M) [4]. Overdose usage of these antibiotics has resulted in the reaching of their residues in surface water, food, soil, etc., particularly in dairy products and other animal-origin foods [2,6,8].

The detection of traces of the contaminants in food is extremely important, as they can have serious health consequences on humans, which range from allergic reactions in sensitive individuals to the evolution of antibiotic-resistant bacteria [2,5,9,10] and enhanced bacterial resistance infections [11].

Traditional methods, such as liquid chromatography-mass spectrometry (LC-MS), electrochemical, and fluorometric methods, have been applied for the detection of *β*-lactam antibiotics or *β*-lactamase inhibitors. Although good results have been obtained from these approaches, they are often time-consuming and require highly expensive equipment, trained personnel, and expensive reagents [12,13].

In recent years, great attempts have been made toward the investment and development of high-throughput assays for the screening of antibiotics, including receptor-based screening for antibiotics residue (i.e., radioreceptor assay, enzyme labeling assays, colloidal gold receptor assay, enzyme colorimetry assay, and biosensor assay). These assays make the detection of antibiotics faster, easier, and more accurate [14]. The use of biosensors is one of the most widely spread techniques applied for the detection of antibiotics and contaminants in foods and environment [14,15,16]. Biosensors are sensing devices that detect contaminants by a combination of a biological sensing element with a detector system while using a transducer [17,18]. They are sensitive, selective, rapid, cost-effective, and portable techniques that replace existing traditional analytical methods [1,19]. In optical biosensors, which are a type of biosensor, the detection of any change in the biological elements is usually dependent on absorption, fluorescence, or light scattering [17,20].

The mechanism of action of *β*-lactam antibiotics (including penicillin G) is based on the inhibition of membrane-associated enzymes that are essentially required to catalyze the final step of bacterial peptidoglycan [5,12,21,22]. Regarding the covalent binding via their active-site serine that is located in the penicillin-binding/transpeptidase domain to penicillin and other *β*-lactam antibiotics [19], these enzymes are called penicillin-binding proteins (PBPs) [23]. There is a differentiation in the number, type, and function of PBPs according to the species of bacteria [14]; for example, *Streptococcus pneumoniae* has three class A PBPs (PBP1a, PBP1b, and PBP2a), two class B PBPs (PBP2x and PBP2b), and one class C PBP (PBP3) with D, D-carboxypeptidase activity. All six PBPs have been implicated in resistance. However, only the two (class B) PBPs perform critical roles in septal and peripheral peptidoglycan synthesis; therefore, they are considered to be primary targets for *β*-lactams [24,25,26].

A variety of radioisotopes with different structural analogs of penicillin have been used in experiments to study the acylation of PBPs [13]. Bocillin-FL [21] is considered to be the most common fluorophore-conjugated analog substrate that has a specific affinity to PBPs, rendering it with a specific fluorescent spectrum. The complex of fluorescent PBP-Bocillin-FL can then be detected using different techniques, such as fluorescence anisotropy, flow cytometry, or UV-Vis spectrophotometry. The indication of losing fluorescence results from the displacement of PBP-bound Bocillin-FL by *β*-lactam antibiotics inhibitors, which can be used as a sensor to measure their concentrations [27].

Real-time polymerase chain reaction (Real-time PCR) is a nucleic acid amplification technology that is based on the detection of fluorescent dye emission attached to the targeted amplicon during the steps of reaction, and the intensity of fluorescence is proportional to the amount of a real-time amplified product; quick results without too much manipulation are usually obtained from this technique [28,29]. A real-time fluorescence detecting instrument was used to identify the Bocillin FL-derivative fluorescent probe of penicillin, depending on the same basis of fluorescent dye detection (present study). The principle of the work in the present study is based on competitive enzyme action on the inducer, which results in an indirect reduction of Bocillin FL intensity with the presence of high concentrations of Penicillin G. These methods can be used as a screening test for the determination of *β*-lactam residues in different samples of food and dairy products.

## 2. Results

Heterologous expression and Purification of PBP2x*

For the purification of the soluble PBP2x (PBP2x*), a PCR of the *pbp*2x gene (2171 bp) was generated with *Nco*I and *Xho*I overhangs, including a deletion in the sequence coding for residues (19–48) located near the N-terminal end of the protein that is responsible for membrane binding [23]. The blunt-end cloning of the PCR fragment into an *EcoR*V digested pUC57 resulted in pUC57-*pbp*2x*. The plasmid was used for the sequence verification of the insert and sub-cloning of the insert into the *Nco*I and *Xho*I cloning sites of the pET28a+ expression vector, creating pET28a+-*pbp*2x* (Figure 1).

### 2.1. Biosensor Assays

The biosensor assays were established based on competition between equal molecules of Bocillin FL (Thermo Fisher Scientific, (Waltham, MA, USA)) and *β*-lactam to carry on an equivalent amount of the purified PBP2x* protein of *S. pnemoniae*.

The correlation between the concentration and the fluorescence intensity rate of Bocillin FL was validated while using a broad range of the compound concentration and later, the test was repeated with a finer concentration range from the compound to confirm the relation. It has been shown that the fluorescence intensity of Bocillin FL was strongly correlated with the used concentrations (0–12 nM) of the compound, and the resulting curve showed good collinearity (0.9781), as shown in Figure 2.

#### 2.1.1. On Column Binding Real-Time Biosensor Assay

This test was established based on the fluorescence intensity of a labeled *β*-lactam analog compound, Bocillin FL. In total, 0.2 nM of Bocillin FL was added to a mixture of 0.25 nM of 6His-tagged PBP2x* with serial concentrations of penicillin G.

The results showed a linear correlation (R^2^ = 0.9825) between the increasing concentration of penicillin G in the reaction mixture (0–0.2 nM) and the amount of the bounded Bocillin FL to the protein, being calculated indirectly by subtracting the fluorescent intensity of the Ni-column wash flow from the entire fluorescence intensity of Bocillin FL (0.2 nM) used in the reaction (Figure 3).

Thermal de-acylation of the antibiotic from the on column PBP2x*-penicillin was performed and the concentration of the eluted free penicillin G was measured by comparison of the HPLC peak area with that of the reference concentration. The accuracy percentage of the biosensor reads was calculated by comparison with that obtained from the HPLC measurement.

A recovery test of the antibiotic in three different samples (honey, milk, and meat-lysate) was performed by adding penicillin G (at concentrations of 0.04, 0.08, 0.16, and 0.2 nM). The later concentration of the antibiotic was quantified while using the on column biosensor method. The highest recovery (97%) was seen in the samples containing 0.2 nM of the antibiotic, followed by the milk sample with the same concentration of the antibiotic. The recovery percentage was lowered by lowering the antibiotic concentration in all three samples. The lowest recovery percentage was found inside the meat-lysate (Figure 4).

The on column real time biosensor assay was used to evaluate hospital-environment contamination with *β*-lactam antibiotics so long as they are specifically attached to the active site of PBP2x* protein. *β*-lactam antibiotics were detected inside environmental samples taken by a smear from the floor of three different rooms of a hospital, a pharmaceutical storage room, and a hospital administration room, as shown in Figure 5. The highest contamination was found in a sample that was collected from the hospital, HFS3, which was a smear taken from the floor of a room of daily drug dosage distribution and patient treatment, followed by a sample, HFS2, which was obtained from the floor of one intensive care room of internal medicine clinic, and the sample HSF1 that was taken from the general entrance of the clinic. The lowest *β*-lactam antibiotic concentration was found in samples that were taken from a hospital administration office, HAOS, and the floor of a pharmacy room, PSFS, of the clinic.

#### 2.1.2. Optimization of Binding of Bocillin FL and PBP2x* for the ChemiDoc-It^®^2 Imager Based Biosensor

We examined 1 µM of Bocillin FL mixed with 1, 0.5, 0.25, and 0.1 µM of soluble PBP2x* after the incubation the signal intensity of the bounded Bocillin FL was measured to optimize the test, as shown in Figure 6A; then, 1 µM of the soluble PBP2x* was examined with 0.5, 0.25, 0.1, and 0.05 µM of Bocillin FL (Figure 6B). Our results showed that using 1 µM of Bocillin FL with 1 µM of soluble PBP2x* was the most stable reaction in all of the repeated experiments.

##### ChemiDoc-It^®^2 Imager as a Detector of Binding Bocillin FL to PBP2x* of S. pneumoniae

Figure 7 shows the calculation curve for the samples with different concentrations of penicillin G. A fixed amount of PBP2x* (1 µM) was added to a varied concentration of penicillin G antibiotics (0.015–2.5 µM) to bind covalently. Later, 1 µM of Bocillin FL was added to the mixture. If no penicillin G was present at the first incubation step, then all of the PBP2x* molecules would be complexed with Bocillin FL; thus, with the presence of a higher concentration of penicillin G in samples in the first incubation, there would be less chance of a PBP2x* complex with Bocillin FL. The solution was subjected to sodium dodecyl sulfate-polyacrylamide gel electrophoresis (SDS-page) and then imaged by a ChemiDoc-It^®^2 Imager biosensor. The total relative intensity of Bocillin FL for each sample was calculated.

Antibiotics are commonly utilized for the treatment of infectious diseases as a means for disease prevention (prophylaxis) in animals. The fast appearance of resistant bacteria is emerging as a potential threat in the next decades, while the extensive use of therapeutic or sub-therapeutic dosages of antibiotics that are used in farms to enhance growth is considered to be important factor that significantly influences the development of bacterial multidrug-resistance [2,5,30]; therefore, consumer and regulatory agencies have a high concern regarding antibiotic residues in food-producing animals.

The development of an appropriate analytical method for the identification and quantification of minimum amounts of antibiotic contaminants in food (e.g., milk, honey) remains important work and it needs to be focused on, as the ubiquitous presence of antibiotics can have serious health consequences on humankind, which range from allergic reactions to the evolution of antibiotic-resistant bacteria [10,31].

In this field, great efforts have been made to develop several adequate and high-throughput confirmatory and screening assays for antibiotics, while biosensors have been widely employed as the most precise, sensitive, cost-effective, and real-time analytical method used for the detection of antibiotic residues [32]

This work presents two related biosensor assays for the detection of penicillin G. The assays were developed on the covalent binding between PBP2x and penicillin G and the penicillin-analogue Bocillin FL in two separate steps. The two assays described are based on the same enzymatic reaction, but they differ concerning the signal detection.

For that, the soluble PBP2x* protein was heterologously expressed as a C-terminal His-tagged fusion derivative in cells of *E. coli* BL21 (DE3) harboring pET28a+-*pbp*2x* plasmid selected with kanamycin 50 µg/mL. The PBP2x* protein carries a deletion in the cytoplasmic membrane binding anchor (19–48 hydrophobic residues) [2]. Protein purification was performed based on the affinity between the histidine tag of the protein and Ni beads. The data from SDS-PAGE electrophoresis were consistent with the expected size of 80.1 kDa for PBP2x*-6His tagged fused protein (Figure 1).

The fluorescence detection method was established with the aid of a real-time PCR machine while using a wavelength of 504 nm for excitation and 511 nm for emission—the same as those used for the excitation of Fluorescein amidites (FAM) dye used for the detection of real-time PCR. The collinearity between the concentration of Bocillin FL and signal intensity confirmed the correct use of the substrate, and the physical condition that was used for the excitation and emission at concentrations varied between 0.02 to 12 nM (correlation coefficients R^2^ = 0.9781) (Figure 2). Later, the PBP2x* protein was incubated with varying concentrations of penicillin G, followed by the next incubation by Bocillin FL. The PBPs that were not complexed by penicillin G in the first incubation step were complexed by Bocillin FL in the second step. Consequently, the intensity of the PBP-Bocillin FL conjugate signal in the reaction mixture was inversely related to the concentrations of the antibiotics in the samples [23], which also showed high co-linearity (correlation coefficients R^2^ = 0.985) while using a real-time PCR machine to read the signals (Figure 3). For further confirmation, the accuracy of this biosensor assay was supported by HPLC analysis by the estimation of the antibiotic eluted from the binding column via thermal de-acylation, which showed an accuracy between 98.4–101.2%. The detection limit was also performed while using SDS-PAGE electrophoresis, and the fluorescence excitation of Bocillin FL was undertaken by a UV Transluminator apparatus with the aid of gel quant software for signal quantification (Figure 6). The results confirmed 0.15 µM as the lowest limit of detection (LOD) of penicillin G using the in-gel quantification method, which is 7500 times higher than the lowest LOD (0.02 nM) of the antibiotic using the real-time PCR machine method (Figure 2 and Figure 3). The sensitivity and accuracy of the method due to the signal reading in a closed system inside the real-time machine protects the fluorophore from the physical damage effects of external light, as well as minimizing the interference of external light in comparison to the open in-gel signal detection. The samples of honey, milk, and meat were supplied with penicillin G at four different concentrations to confirm whether the on column binding real-time method can be adapted to the quantification of penicillin G traces inside biological samples. The highest recovery was found in a honey sample that was supplied with 0.2 nM of penicillin G, while the lowest recovery (less than 50%) was seen in a meat sample that was supplemented with 0.04 nM of the antibiotic. Although defatted milk and centrifuged meat were used in the assay, the protein content of the samples might interfere with the binding of both the antibiotic and Bocillin FL to the binding site of the PBP2x protein in comparison to the honey sample. Overall, more than 80% recovery was seen in the samples when supplemented with the antibiotic concentrations 0.2, 0.16, and 0.12 nM, except the sample of honey, which contained 0.16 nM of the antibiotic.

We assume that the method will be more reliable at concentrations greater than 0.16 nM of the antibiotic inside the biological samples, and deproteinization treatment or dilution of the samples before testing can greatly approve the recovery percentage, although the sensitivity of the method is satisfactory to detect penicillin below the European Maximum Residue Limit (MRL) [2,7].

The effluent of antibiotics onto the surface area and subsequent leakage into hospital sewage water after the prolonged treatment of patients, particularly with broader-spectrum antibiotics, contribute to the contaminated sources of emerging drug-resistant microbes [33].

In this work, different smear samples from hospital floors were explored for penicillin G traces using the established biosensor assay with the aid of a real-time PCR machine to read the fluorescence signal and, interestingly, all of the samples were positive for the antibiotic detection. The maximum antibiotic contamination was detected in a sample taken from the floor of a patient treatment room, followed by a sample taken from an intensive care unit (0.27 nM, and 0.13 nM, respectively), and the lowest contamination was found in a sample taken from floor of an administration room of the hospital (0.014 nM) (Figure 5). It has been confirmed that most antibiotics are not completely metabolized in the human body, and studies confirmed the excretion of about 25–75% of given antibiotics to the environment in an inactive form through coughing, vomiting, urine, and feces; thus, antibiotics that are used by a humans can be found at various concentrations in hospital effluents [34,35]. In addition, the use of processed and untreated wastewater as fertilizer might contribute to a significant contamination and development of antibiotic-resistant bacteria [36,37]. Although some antibiotics in nature can degrade very fast, some of them can persist for a long time [36]; however, the principal of degradation mechanism of each antibiotic basically relies on its chemical structure [38]. Contrary to expectations, low contamination was noted in the samples that were taken from a pharmacy store in comparison to the other areas of the clinic, except for the administration room. This can be explained by extra care from the pharmacist during the antibiotic distribution and preventing damage to the packaging material of the antibiotics. Besides, professional and deep cleaning of the area by the hospital service staff has a significant impact on this type of contamination.

To the best of our knowledge, this is the first report regarding developing an ultra-sensitive nanoscale biosensor system for the screening and real-time measurement of penicillin G antibiotic with an improved signal detection method using real-time PCR assay in comparison to previously published biosensors [6,7,21,23]. Furthermore, this method can be further developed due to the demand for a simple and highly selective low-cost assay for the pharmacokinetic study of antibiotics in different human body fluids. Furthermore, the method can potentially be used for the detection of the of amount disposed-of penicillin G from the production sites, cleaning validation, and inside industrial wastewater flow. Environmental monitoring has been one of the priorities on a global scale, due to the close relationship between environmental pollution by toxic elements and human health/socioeconomic development.

## 3. Materials and Methods

The chemicals used in this work were of analytical grade. The working dilution of penicillin G was freshly prepared using 10 mM potassium phosphate buffer, pH 7.2/50 mM NaCl.

### 3.1. Expression and Purification of Soluble PBP 2x*

The wild-type *Streptococcus pneumonia* (strain AAF17263.1)-*pbp*2x gene was amplified by PCR using Phusion DNA Polymerase and specific primers; PBP-up: Primer with *Nco*I overhang ACCATGGTGAAGTGGACAAAAAGAGTAATCCGTTATG CGACCAA

AAATCGGAAATCGCCGGGACAGGCACTCGCTTTGG and PBP-His-*Xho*I-dn: Primer with *Xho*I overhang CTCGAGGTCTCCTAAAGTTAATG to generate a blunt-ended (*pbp*2x* 2172 bp) fragment with *Nco*I and *Xho*I overhangs, carrying a deletion near the five-end sequence of the gene coding for membrane binding domain. This fragment was first ligated as a blunt end into the pUC57 vector, which was previously digested by the *Eco*RV enzyme (Fermentas, Germany). The construct was transformed into competent cells of *E.coli* DH10B. A correct transformant was grown on luria bertani (LB) agar that was supplemented with 100 mg/mL ampicillin. Subsequently, the resulting vector (pUC57-*pbp*2x*) was isolated and digested using the restriction enzymes *NcoI* and *XhoI*, and the insert was ligated into a linearized pET-28a+ plasmid vector that was previously double digested with *Nco*I and *Xho*I restriction enzymes (Fermentas, Germany); the ligation reaction was transformed first into cells of *E. coli* DH10B, and correct transformants with pET-28a+−*pbp*2x* were selected on LB agar plates that were supplemented with 50 mg/mL kanamycin. The expression vector coding for soluble PBP2x* with a C-terminal 6His-tag was checked by the restriction enzymes and further confirmed by the sequence analysis of the insert. The correct plasmid was transformed into the cells of *E. coli* BL21 for heterologous expression.

The plasmid was grown until OD600 nm = 0.6, and then IPTG (0.01 mM) was added to induce the protein expression in order to obtain soluble PBP2x* with a His-tag culture of *E. coli* BL21; then, the cells were harvested and disrupted by an ultrasonic homogenizer. Soluble PBP2x* was purified from the supernatant while using the His•Bind^®^/Ni^−^ column (Roth, Germany), following the manufacturer’s instructions, and analyzed on SDS-10% PAGE with Coomassie Brilliant Blue staining. The protein concentration was determined by the Bradford method and stored at −80 °C in the presence of 10% glycerol for further use.

### 3.2. Biosensor Assay for Penicillin G Detection

#### 3.2.1. On Column Binding Real-Time Biosensor Assay

The assay was based on competition between an equal number of molecules from Bocillin FL (Thermo Fisher Scientific, (Waltham, MA, USA)) and *β*-lactam on binding to an equivalent concentration of PBP2x* protein of *S. pneumoniae*.

First, finding and verifying a method was required in order to estimate free and bounded Bocillin FL by the excitation of the material at 504 nm and measuring produced light at 511 nm. For that, a real-time on column based method was used as a tool for the quantification of Bocillin FL. Verification of the method was performed to confirm the relationship between Bocillin FL concentration and fluorescence light intensity while using first coarse concentrations and, later, the test was repeated using finer concentrations that ranged from 0 nM to 0.25 nM of the compound, in accordance with the compound concentrations that were used in the proceeding experiments.

Later, penicillin G was mixed into a binding buffer containing 10 mM potassium phosphate pH 7.2 with 50 mM NaCl, 0.25 nM His-tagged soluble PBP2x* (concentration varied from 0–0.25 nM), and incubated for 15 min. at room temperature for binding, followed by the addition of Bocillin FL to occupy the remaining free sites of the protein, and the mixture was again incubated for another 15 min. Later, the mixture was loaded onto a column containing Roti^®^garose Ni beads that were pretreated with binding buffer (50 mM Tris-HCl (pH 7.9), 200 mM NaCl, and 10% glycerol) to capture the PBP2x*–penicillin–Bocillin FL complex through the histidine residues of the protein. After washing the column using washing buffer, the non-bounded free Bocillin FL inside the flow-through and eluted Bocillin FL within the complex PBP2x*–ampicillin–Bocillin FL were measured by the use of a real-time PCR apparatus at a wavelength of 511 nm emission upon excitation at 504 nm, as described previously. The concentration of Bocillin FL was calculated from the standard curve that was formerly prepared for this purpose.

#### 3.2.2. HPLC Analysis

HPLC analysis (C18 column, 10 cm × 4.6 mm, I.D. 2.7 μm particle size) was performed by the injection of 10 µL of 100× concentrated sample, using mobile phase (A): 10 mM ammonium acetate (pH 4.5 acetic acid) and (B): acetonitril (75:25, A:B) 1 mL/min isocratic for 8 min. at UV 220 nm using Ultimate3000 Thermo Scientific HPLC/UPLC Chromeleon (C) Dionex (Version 7.2.9.11323).

#### 3.2.3. Recovery Test and Method Validation:

A recovery test was executed to confirm the sensitivity and accuracy of on-column binding real-time based quantification assay using biological samples (honey, defatted milk, and meat-lysate). Different concentrations of penicillin (final concentrations of 0.04, 0.08, 0.12, 0.16, and 0.2 nM) were added to 100 µL of the samples; then, 100 µL of 2× binding buffer was added and mixed well. The mixture was added to His-tagged soluble PBP2x* (0.25 nM final concentration) and incubated for 15 min. at room temperature. The mixture volume was completed to 1 mL by adding the binding buffer and loaded onto a Roti^®^garose Ni beads column and centrifuged briefly. Then, 0.25 nM of Bocillin FL in 200 µL buffer was added to the Ni-beads and incubated for 15 min. at room temperature to bind to the protein. Subsequently, the column flow-through was collected and fluorescence intensity was calculated, as formerly described.

#### 3.2.4. Quantification of Penicillin G in the Hospital Environment

Smears from the floors of working areas from three hospitals, a pharmaceutical storage room, and a hospital administration room were collected using sterilized cotton swabs. The swabs were impregnated in 200 µL of the binding buffer and stored at −20 °C until use. The quantification of penicillin G antibiotic in the samples was performed using the previously described on-Ni-column binding real-time quantification assay.

#### 3.2.5. ChemiDoc-It^®^2 Imagers- Based Quantification Assay

The assay was based on the inhibition binding of soluble PBP2x* of *S. pneumoniae* to a fluorescently labeled *β*-lactam analogue (Bocillin FL). Here, different concentrations (0.015 to 2.5 µM) of penicillin G were incubated with 1 µM of soluble 6X-Histidine tagged PBP2x* for 15 min. at room temperature [23], whereby penicillin G would bind to the PBP2x*. Non-bounded 6His-soluble PBP2x* was then allowed to interact with 1 µM Bocillin FL at room temperature, followed by adding 6 µL of loading buffer (without dye) to each reaction. The samples were separated by electrophoresis on SDS-polyacrylamide gels (6% stacking gel and 10% separating gel) at 120 V, for 4 h, and Roti-Mark 10-150 Plus (Roth, Germany) was used as a molecular marker [39]. The SDS-gel was then imaged by a ChemiDoc-It^®^2 Imager (Transluminator UV Model 344-100, USA) and the band concentrations were analyzed based on Gel-quant software.

## 4. Conclusions

In this study, we developed a rapid and sensitive on-column real time method for the quantitation of *β*-lactam residues using a receptor protein (PBP2x*) of *Streptococcus pneumoniae* for the first time. The on-column real-time methodology measured concentrations of antibiotics of higher quality in comparison to the SDS-based system approach. The highest recovery of penicillin G was found in the honey sample, whereas the lowest recovery (less than 50%) was observed in the meat-lysate sample. The results indicate that the developed assay could emerge into a screening assay for routine use.

## Figures and Tables

**Figure 1 molecules-25-01248-f001:**
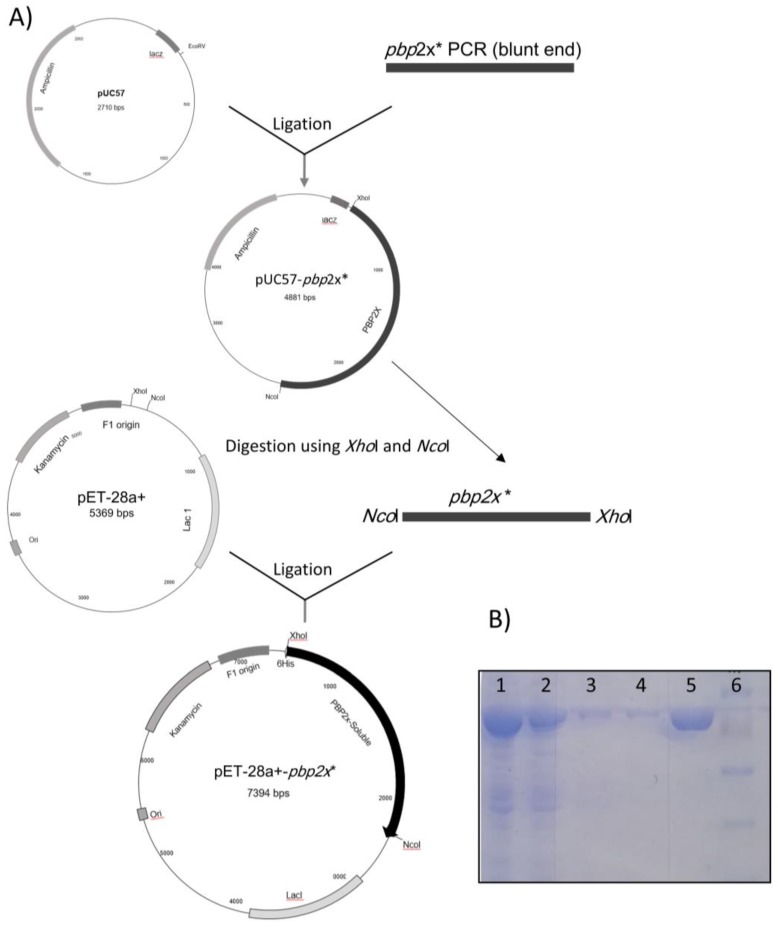
(**A**) Cloning of *pbp*2x* (2171 bp) PCR (blunt end) into the multiple cloning sites (*EcoR*V) of the pUC57 vector resulted in the production of pUC57-*pbp*2x* (4881bp). The insert is sub-cloned into the *Nco*I and *Xho*I sites of the predigested pET28a+ expression vector creating pET28a + −*pbp*2x*. (**B**) SDS-page shows the expression of the soluble PBP2x* protein in cells of *E. coli* BL21, lane 1: Whole crude extract of *E. coli* BL21-pET28a + −*pbp*2x* after induction with Isopropyl *β*-d-1-thiogalactopyranoside (IPTG), lane 2: flow-through using 40 mM imidazole-containing buffer, lane 3: flow-through using 60 mM imidazole-containing buffer, lane 4: purified PBP2x* (80.1 kDa) eluted from the purification column using 200 mM imidazole, and lane 5: pre-stained protein marker (10–150 kDa).

**Figure 2 molecules-25-01248-f002:**
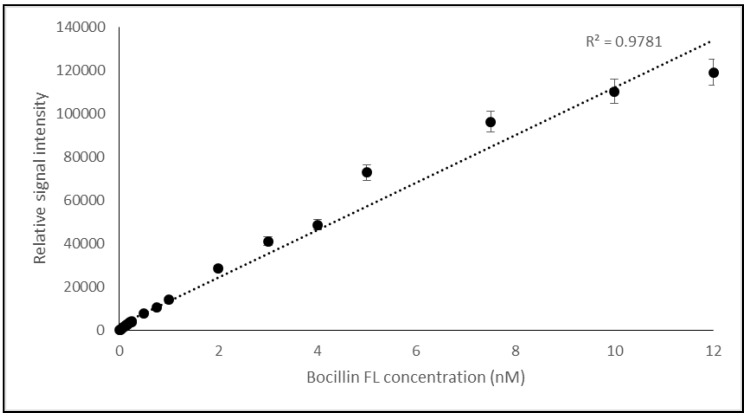
Calibration curve of the Bocillin FL fluorescence intensity using concentrations varying from 0–12 nM of the compound while using a real-time PCR machine for reading (at wavelengths of 511 nm for emission and 504 nm for excitation), which shows the collinearity between concentration and signal intensity (R^2^ = 0.9781).

**Figure 3 molecules-25-01248-f003:**
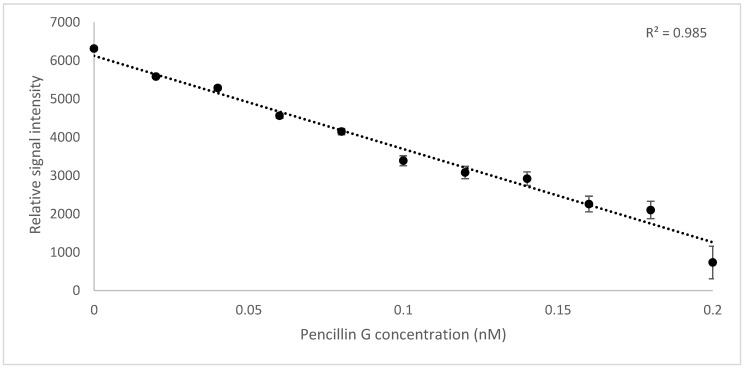
Penicillin quantification based on the inhibition of on-column Bocillin FL binding to PBP2x protein by the addition of gradient concentrations of the antibiotic (0–0.2 nM) to 0.2 nM of the protein. The curve shows the collinearity (R^2^ = 0.985) between the antibiotic concentration and the fluorescence intensity of bounded Bocillin FL to the protein, measured by real-time PCR machine (excitation at wavelength 504 nm and emission at 511 nm).

**Figure 4 molecules-25-01248-f004:**
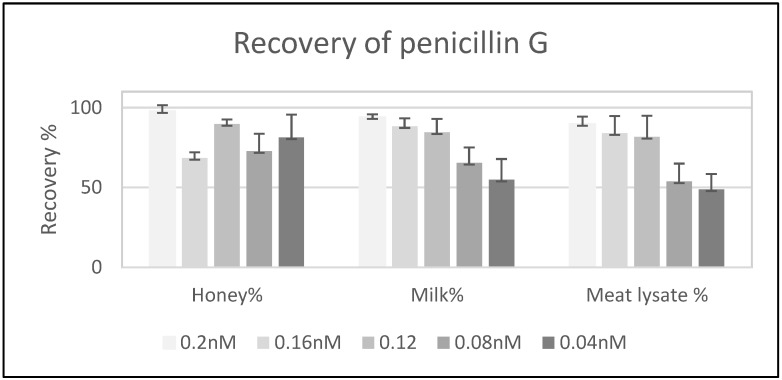
Recovery percentages of penicillin G in different matrixes. Concentrations (0.04 nM, 0.08 nM, 0.12 nM, 0.16 nM, and 0.2 nM) of the antibiotic added to samples of honey (first column group), defatted milk (second column group), and meat lysate (Third column group). The recovery (%) is calculated based on the indirect measurement of the bounded antibiotic to 6His-PBP2x* tagged protein using real-time biosensor assay.

**Figure 5 molecules-25-01248-f005:**
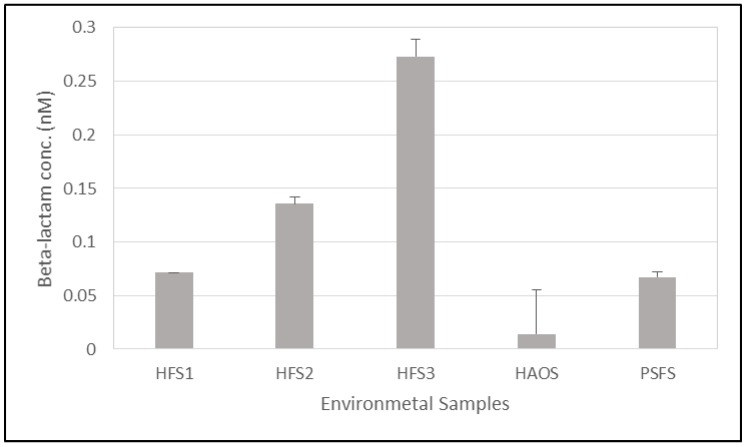
Quantification of penicillin G traces in five contaminated environmental samples (floor smear) including the clinic general entrance (HFS1), intensive care (HFS2), daily drug distribution and patient treatment room (HSF3), hospital administration office (HAOS), and the pharmaceutical storage of the clinic (PSFS).

**Figure 6 molecules-25-01248-f006:**
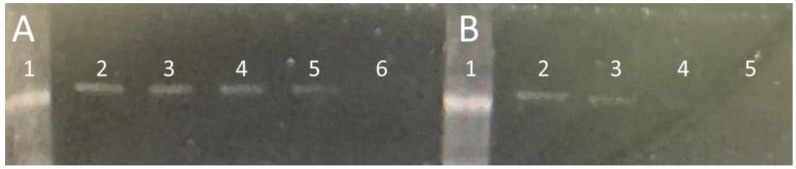
SDS gage for optimization of binding of Bocillin FL and PBP2x* imaged ChemiDoc-It^®^2 Imager. (**A**) Lane 1: protein marker 70 kDa band (PageRuler Pre-stained NIR Protein Ladder (Thermo Fisher Scientific (Waltham, MA, USA)), lanes 2-6: Binding reaction of 1 µM of Bocillin FL with 1, 0.5, 0.25, and 0.1 µM of soluble PBP2x*, respectively. (**B**) Lane 1: Protein marker, lanes 2-5: Binding reaction of 1 µM of soluble PBP2x* with 0.5, 0.25, 0.1, and 0.05 µM of Bocillin FL, respectively.

**Figure 7 molecules-25-01248-f007:**
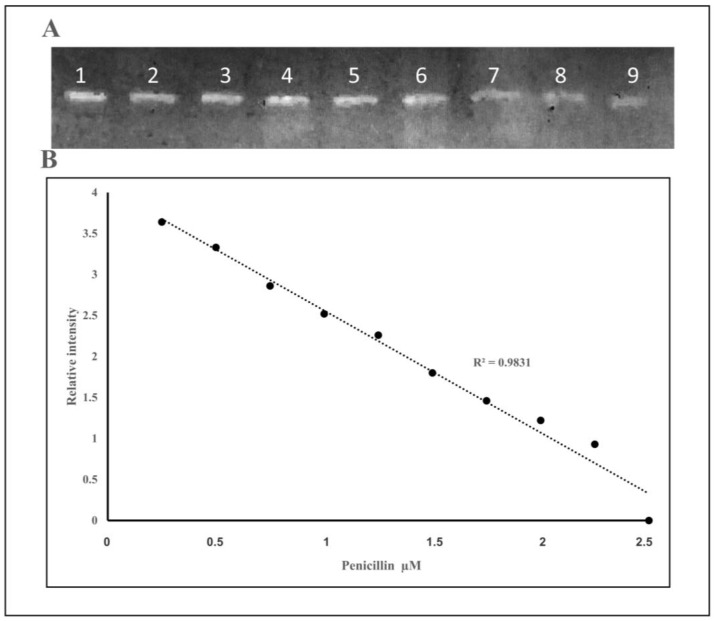
(**A**) Lanes (1–9): SDS-page of the binding reaction of 1 µM soluble PBP2x* with 0.15 to 2.5 µM of Penicillin G, followed by addition of 1 µM of Bocillin FL (imaged by ChemiDoc-It^®^2). (**B**) Correlation between relative fluorescence intensity of Bocillin FL and penicillin G concentrations quantified by Gel-quant software from the image A.

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
