# Peer review of "On Column Binding a Real-Time Biosensor for β-lactam Antibiotics Quantification"

_molecules, 2020, doi:10.3390/molecules25051248_

Round 1

Reviewer 1 Report

Biosensors are sensitive and selective techniques that can replace the existing instrumental methods in many applications. Thus, the present paper on developing an ultra-sensitive biosensor system for the screening of beta-lactam antibiotics could be interesting for a reasonable number of scientists.

The manuscript is well organized and written. However, the paper could be improved. Some editorial changes could be included, e.g.: the capital letters in some words, mg/ml and µg ml-1, etc.

Scientific novelty is clearly showed. However, more comments could be added in some sections, e.g.:

“Although some antibiotics in nature can degrade very fast, some of them can persist for a long time (37).” It is an interesting problem and needs more explanation.

“Honey, milk, and meat extract”, what does it mean?

Author Response

Reviewer 1

Point1: Some editorial changes could be included, e.g.: the capital letters in some words, mg/ml and µg ml-1, etc.

Response 1: According to the reviewer's recommendation, all the editorial changes are done. The manuscript is again reviewed by MDPI English editing and proof certificate uploaded.

Point 2: “Although some antibiotics in nature can degrade very fast, some of them can persist for a long time (37).” It is an interesting problem and needs more explanation.

Response 2: According to the reviewer's recommendation, more explanation added to the text according to the reviewer’s recommendation (Line 266 to 270)

Point 3: “Honey, milk, and meat extract”, what does it mean?

Response 3: According to the reviewer’s note:

  • The word extract is removed, and meat extract changed to meat lysate as in the lines 142, 148, 152, 346, and 377

Reviewer 2 Report

Comments

The author essentially used a RT-PCR machine for fluorescence reading, instead of using a fluorometer. There is no PCR involved, and thus it is incorrect to call the assay real-time PCR-based. The authors did not show the real-time aspect of the assay either, as each sample contain a single concentration of analyte that doesn't change over time. For these reasons, it is also incorrect to call the assay a real-time biosensor.

Figure 1 shows the fluorescence of Bocillin FL that is not bound to PBP2x. What would be the fluorescence signal when Bocillin FL is bound to PBP2x as in the assay? According to Line 131 in the text, fluorescence signals were measured from the imidazole elution of all bound PBP2x proteins (i.e., Bocillin FL-bound PBP2x). In this case, if there is significant difference in fluorescence when Bocillin FL is bound to PBP2x, the quantification in this work will not be accurate. In addition, there should be an inset showing the calibration curve from 0 to 0.2 nM, because the assay results were derived from the competition of the antibiotic with 0.2 nM Bocillin FL. These data are essential to support the validity of the assay, and the authors should clarify how the assays were performed.

The presence of Penicillin G should decrease the amount of Bocillin FL bound to PBP2x, hence a decrease in fluorescence signal would be expected with increasing Penicillin G concentrations. Why in Figure 3, the fluorescence signal increases (i.e., increase of bound Bocillin FL) with Penicillin G concentration? The caption clearly states that the signal was that of the PBP2x-bound Bocillin FL. The data presented in Figure 3 appear to be in contradiction to what the assay is expected to behave.

Different beta-lactam antibiotics would have different affinity to PBP2x and other binding characteristics. Due to these differences, a calibration curve has to be generated for each specific antibiotic. Without these experiments, the quantification of total beta-lactam antibiotics in Figure 5 cannot be considered valid.

Minor comments:

Editing of the English text may be required “l-1” not properly written, “-1”should be superscript A number of errors have to be corrected, for example “nm of Bocillin” should be “nM of Bocillin FL” Formatting for lines 95 to 100

Recommendation

Due to the lack of essential experiments/data required to support the performance of the assay, as well as the confusing data presented in Figure 3, the manuscript cannot be recommended for publication in its present form.

Author Response

Reviewer 2

Comments

Point1: The author essentially used a RT-PCR machine for fluorescence reading, instead of using a fluorometer. There is no PCR involved, and thus it is incorrect to call the assay real-time PCR-based. The authors did not show the real-time aspect of the assay either, as each sample contain a single concentration of analyte that doesn't change over time. For these reasons, it is also incorrect to call the assay a real-time biosensor.

Response 1: We agree with the reviewer that the RT-PCR machine is not used for PCR generation, however, the method is established based on a fast and real-time reading of the fluorescence flow which represents the antibiotic concentration in comparison to the time-consuming SDS gel electrophoresis image scanning for the signal quantification. For that the method used in this work is named real-time biosensor assay, we hope the reviewer accepts our explanation.

Point 2: Figure 1 shows the fluorescence of Bocillin FL that is not bound to PBP2x. What would be the fluorescence signal when Bocillin FL is bound to PBP2x as in the assay? According to Line 131 in the text, fluorescence signals were measured from the imidazole elution of all bound PBP2x proteins (i.e., Bocillin FL-bound PBP2x). In this case, if there is significant difference in fluorescence when Bocillin FL is bound to PBP2x, the quantification in this work will not be accurate. In addition, there should be an inset showing the calibration curve from 0 to 0.2 nM, because the assay results were derived from the competition of the antibiotic with 0.2 nM Bocillin FL. These data are essential to support the validity of the assay, and the authors should clarify how the assays were performed.

Response 2:

The first assay was performed to confirm the relationship between free Bocillin FL concentrations and fluorescence intensity estimated by the real-time PCR machine without involving the PBP2x protein. In the next assay, we showed the correlation between the signal intensity of the non-bounded Bocillin FL which resulted from the competition of the antibiotic concentration with 0.2 nM Bocillin FL, and the result showed a linear (reverse) relationship between the antibiotic concentrations and the fluorescence read.

Furthermore, upon request of another reviewer, we have recently confirmed this correlation by HPLC analysis of the free antibiotic done after a thermal de-acylation of the antibiotic as explained in the line (138 -142) , (231-232), and (338-342).

Point 3: The presence of Penicillin G should decrease the amount of Bocillin FL bound to PBP2x, hence a decrease in fluorescence signal would be expected with increasing Penicillin G concentrations. Why in Figure 3, the fluorescence signal increases (i.e., increase of bound Bocillin FL) with Penicillin G concentration? The caption clearly states that the signal was that of the PBP2x-bound Bocillin FL. The data presented in Figure 3 appear to be in contradiction to what the assay is expected to behave.

Response 3: The experiment represents a linear (reverse) relationship between the antibiotic concentrations and the fluorescence read as also explained in the response for the reviewer’s note 2.  Higher the concentration of the non-bounded Bocillin FL in the column flow represents higher concentration of penicillin G that complexed with the PBP2x protein (Figure 3), while in the gel method estimation, higher Bocillin FL concentration reflects less penicillin G concentration complexed with the PBP2x protein (Figure 7).     

Point 4: Different beta-lactam antibiotics would have different affinity to PBP2x and other binding characteristics. Due to these differences, a calibration curve has to be generated for each specific antibiotic. Without these experiments, the quantification of total beta-lactam antibiotics in Figure 5 cannot be considered valid.

Response 4: We agree with the reviewer. In this work, the assay is established for penicillin G. Also the calibration curve is done for penicillin G not for all beta-lactams, although the established method can be adapted for all kinds of the beta-lactams and cephalosporins. 

According to the reviewer’s note we have changed the word beta-lactam to penicillin G.   

Minor comments:

Point 5: Editing of the English text may be required “l-1” not properly written, “-1”should be superscript A number of errors have to be corrected, for example “nm of Bocillin” should be “nM of Bocillin FL” Formatting for lines 95 to 100.

Response 5: Editing of the English text has been done, the the manuscript is completely subjected to the MDPI English Editing, and the proof certificate is uploaded.

Recommendation

Due to the lack of essential experiments/data required to support the performance of the assay, as well as the confusing data presented in Figure 3, the manuscript cannot be recommended for publication in its present form.

Response: Figure 3 and the assay result (Linear reverse correlation between Bocillin FL signal and penicillin concentration) are well explained for the reviewer (in answer no. 2 and 3). Furthermore, to support our data we have newly performed HPLC analysis for the antibiotic after a thermal de-acylation of the antibiotic as explained in the line (138 -142) , (231-232), and (338-342).

Reviewer 3 Report

There are many beta-lactam antibiotics used in clinical area. Some issues should be concerned.

1. How to distinguish penicillin G with other beta-lactam antibiotics?

2. Please use other method to confirm the results of proposed method are correct, such as HPLC method or GC method.

3. Figures 2 and 3 should add the error bars. 

Author Response

Reviewer 3

Point 1: How to distinguish penicillin G with other beta-lactam antibiotics?

Response 1: Although PBP2X shares the same binding site for binding to all beta-lactam antibiotics, however this assay does not distinguish between the beta-lactam antibiotics, therefore, the current study is specified for penicillin G estimation. Accordingly, the word beta-lactam in the manuscript is replaced by penicillin G.   

Point 2: Please use other method to confirm the results of proposed method are correct, such as HPLC method or GC method.

Response 2: We agree with the comment, for that, the biosensor assay of Penicillin G concentration was repeated and then thermal de-acylation of the antibiotic of PBP2x-Beta-lactam complex (paper) was performed, followed by HPLC analysis of the eluted free penicillin G. The result was concordant with that estimated by the biosensor. The procedure and results added to the manuscript as in thes line (138 -142) , (231-232), and (338-342).

Point 2: Figures 2 and 3 should add the error bars. 

Response 3: We agree with the reviewer's recommendation, and accordingly error bars added to the figures 2 and 3.  

Round 2

Reviewer 2 Report

This reviewer insists that, using a PCR machine for signal readout cannot make the assay a PCR-based method, there is no PCR involved in the assay. The assay is referred as PCR-based in the abstract, section title, figure caption, etc, multiple times. This is incorrect.

As indicated in this reviewer's original review, Figure 3 caption clearly states that the signal intensity was that of the "fluorescence intensity of bounded Bocillin FL to the protein". If the caption is correct, this signal should not increase as the amount of the added Pencillin G increases. On the other hand, in the author's response, it seems that the authors refer the signal in Figure 3 to the flowed through, non-bound Bocillin FL, which is in contradiction to the figure caption in the revised manuscript. This is the key result of the work and it needs to be thoroughly clarified. However, it is not clear in the experimental method how the data in Figure 3 was generated and clarification and correction are needed.

Due to the incorrect description of the assay, as well as the confusion related to the data presented in Figure 3, this reviewer cannot recommend acceptance of the revised manuscript for publication.

Author Response

- The English language improved, with spell-checking for the whole manuscript according to the reviewer's recommendations

- Research design, methods, and presenting the results including the
figures which are improved and updated according to the reviewer's recommendations.

- Regarding the use of PCR machine, it's used for signal readout, according to the recommendations from the reviewer it's corrected from PCR-based method to ''on-column binding real time biosensor'' and the PCR-based method deleted inside the whole manuscript (Title, abstract, results, figures, and discussion. 

- Regarding the figure 3, the figure is updated, data re-calculated by subtracting the fluorescent intensity of the Ni-column wash flow from the entire fluorescent intensity of Bocillin FL (zero Penicillin). Figure caption is updated based on the reviewer's recommendation.  

Reviewer 3 Report

No extra comment.

Author Response

According to the review's recommendation the introduction, research design, method, results including the figures, and the conclusion parts are improved. 

In addition, English language proofreading, and spell checking done for the whole manuscript.